# Subthreshold Micropulse Laser for Diabetic Macular Edema: A Review

**DOI:** 10.3390/jcm12010274

**Published:** 2022-12-29

**Authors:** Barbara Sabal, Sławomir Teper, Edward Wylęgała

**Affiliations:** 1Chair and Clinical Department of Ophthalmology, Faculty of Medical Sciences in Zabrze, Medical University of Silesia, Okregowy Szpital Kolejowy, Panewnicka 65, 40-760 Katowice, Poland; 2Department of Ophthalmology, John Paul II Municipal Hospital, 35-241 Rzeszow, Poland

**Keywords:** subthreshold laser, micropulse laser, diabetic macular edema, combined treatment

## Abstract

Diabetic macular edema (DME) is one of the main causes of visual impairment in patients of working age. DME occurs in 4% of patients at all stages of diabetic retinopathy. Using a subthreshold micropulse laser is an alternative or adjuvant treatment of DME. Micropulse technology demonstrates a high safety profile by selectively targeting the retinal pigment epithelium. There are no standardized protocols for micropulse treatment, however, a 577 nm laser application over the entire macula using a 200 μm retinal spot, 200 ms pulse duration, 400 mW power, and 5% duty cycle is a cost-effective, noninvasive, and safe therapy in mild and moderate macular edemas with retinal thickness below 400 μm. Micropulse lasers, as an addition to the current gold-standard treatment for DME, i.e., anti-vascular endothelial growth factor (anti-VEGF), stabilize the anatomic and functional retinal parameters 3 months after the procedure and reduce the number of required injections per year. This paper discusses the published literature on the safety and application of subthreshold micropulse lasers in DME and compares them with intravitreal anti-VEGF or steroid therapies and conventional grid laser photocoagulation. Only English peer-reviewed articles reporting research within the years 2010–2022 were included.

## 1. Introduction

Diabetes mellitus (DM) has become a civilization disease associated with a sedentary lifestyle and the aging of the population in the contemporary world. It is estimated that DM affects around 10% of the global population [1]. The prevalence of diabetes is increasing rapidly, and the World Health Organization (WHO) has recognized diabetes as a noncommunicable disease which is causing an epidemic in the 21st century. An insufficiently controlled and long-term disease is associated with a high risk of multiorgan complications, including those involving eyes. One of the main retinal complications is diabetic macular edema (DME), which leads to gradual visual impairment, especially at working ages. DME occurs in 4% of patients diagnosed with DM, even at the early stage of diabetic retinopathy. The estimated number of adults worldwide in 2020 with clinically significant DME was 18.8 million, and this is projected to increase by half in 2045 [2].

According to the current international guidelines for the management of DME by the European Society of Retina Specialists (EURETINA), intravitreal anti-vascular endothelial growth factor (anti-VEGF) was established as first-line therapy in DME with visual impairment [3]. After publishing the results of the DRCR.net Protocol I and Protocol S studies, laser therapy was regarded as being inferior to anti-VEGF treatments [4].

The availability of anti-VEGF injections has changed the standard of care for DME patients [5]. The agents improve both the functional and anatomical parameters of the retina. Currently, different anti-VEGF agents such as ranibizumab, aflibercept, brolucizumab, faricimab, and off-label bevacizumab have become the therapy of choice in DME treatments [6]. Intravitreal injections have a high efficacy and safety profile, however, on the other hand, the therapy is expensive, needs to be repeated, and can very rarely lead to endophthalmitis [7]. To preserve the therapeutic effect, frequent visits and injections are required [8].

As a second line treatment, in the case of chronic, persistent edema, intravitreal steroid administration can be performed. The therapy was approved for a DME that is resistant to anti-VEGF treatment and is preferred for ineligible patients (e.g., those who experienced cardiovascular events in the last 6 months, pregnancy, and hypersensitivity) [8]. A biodegradable intravitreal implant with dexamethasone was approved for use in DME therapy. It releases the steroid for up to 6 months in the vitreous cavity. The clinically relevant adverse events are a secondary cataract formation and elevated intraocular pressure [9].

The previous standard therapy for clinically significant DME was conventional retinal laser photocoagulation. A focal treatment treats areas of focal edema and leakage, whereas a grid-pattern treatment is applied for diffuse macular edemas [10]. According to the Early Treatment Diabetic Retinopathy Study (ETDRS), focal argon photocoagulation with a visible burn after a 3 year follow-up reduced moderate vision loss by half in clinically significant DMEs [11]. The treatment was shown to be effective, but it caused the destruction of the photoreceptors, choroidal neovascularization, and the secondary proliferation of retinal Müller glia, resulting in epiretinal membrane formation [12]. The complications included central scotomas, the deterioration of color and night vision, contrast sensitivity decrease, accidental foveal burn, and laser scar enlargement [3]. With respect to additional disadvantages, the procedure is painful and can induce elevated ocular pressure [10]. Due to its short-term and long-term adverse effects, photocoagulation is no longer the gold standard treatment. Current relative indications for conventional laser treatment are vasogenic DMEs with focally leaking capillaries, DMEs below 300 μm, and persisting vitreomacular adhesion, as a second line treatment in resistant and non-fovea-involving edemas [3].

The subthreshold micropulse laser is a relatively new technology based on cell photostimulation that enables us to decrease the total laser energy [13]. Contrary to classic retinal photocoagulation, micropulse laser treatment (MPLT) is a safe, nondamaging therapeutic tool, selectively targeting the retinal pigment epithelium (RPE) which minimizes the amount of chorioretinal scarring. Laser therapy shows slower onset at the third month after the treatment, with a longer effect on the retina [14]. The series of short repetitive laser impulses enables the tissue to cool down its temperature to avoid thermal burns. A duty cycle (DC) is described as the effective duration of the laser work and it is usually set from 5% to 15%, which means that a single impulse has a duration of 100–300 μs which is interposed by a 1700–1900 μs interval without energy transmission [15]. There are no specified parameters for the MPLT, and various settings for the spot diameter, pulse duration, power, and number of delivered spots have been proposed. The power level of laser can be fixed or set according to the titration over the non-edematous area of the peripheral retina [16]. The spots should be applied in a non-spaced manner over the entire macular area between the vascular arcades, covering the edematous retina and the foveal center. The greatest difficulty in performing MPLT is the invisibility of the laser spots, thus, it is hard to confirm the proper performance of the procedure. A confluent and extensive treatment for the macula is essential to obtain proper results. Multi-spot systems allow is to deliver spots in a regular pattern, e.g., a 7 × 7 matrix, which reduces the time necessary for the treatment, eases the application, and makes it more reliable [17]. Moreover, it is easy to learn for ophthalmologists and, on the other hand, it is safe, comfortable, and painless for the patients. The procedure can be repeated; the retina remains intact and there is no visible scar.

The micropulse treatment can be performed with 532 nm (green), 577 nm (yellow), 810 nm (infrared), or novel 670 nm (red) wavelength lasers. The first clinical application of the micropulse was performed with a 810 nm laser modality, where the wavelength deeply penetrated the retina and, thus, it was not absorbed by the macular carotenoids. The 577 nm wavelength targets oxyhemoglobin and melanin, and it is not absorbed by xanthophyll in the neurosensory retina. Commercially available devices can deliver conventional and micropulse shots at 577 nm that simultaneously enable the combination therapy of the grid micropulse laser and the direct photocoagulation of microaneurysms [18]. The 670 nm laser is less scattered and not absorbed by hemoglobin and xanthophyll, thus, it seems to be safe for the neurosensory retina [19]. There is no consensus on which wavelength is the most favorable for the treatment for DMEs, thus, all the above-described devices have a high safety profile and are recommended for micropulse use.

Currently, the specific indications for the application of MPLT have not been established. It is considered as an alternative treatment in macular disorders such as DME (Figure 1), central serous chorioretinopathy, and macular edemas that are secondary to retinal vein occlusion [20,21,22,23]. MPLT was proven to be efficient and free from adverse events in minor and moderate macular edemas with a central retinal thickness (CRT) below 400 μm and relatively good visual acuity [24]. As an adjuvant to anti-VEGF agents, it helps to stabilize the anatomic and functional retinal parameters with a lower required number of injections.

## 2. Materials and Methods

The present paper reviews all the relevant literature on DME treatments with a subthreshold micropulse laser. The PubMed database and Mendeley were used as a source of studies within the years 2010–2022. Only peer-reviewed articles published in English reporting research were included. Relevant studies were identified using the following terms in combination with Boolean operators: subthreshold laser, micropulse laser, diabetic macular edema, clinically significant macular edema, anti-VEGF, intravitreal steroid, vitrectomy, conventional photocoagulation, ETDRS photocoagulation, continuous-wave photocoagulation, combined therapy, and safety. Subsequently, a manual search of the reference lists in the retrieved manuscripts was performed. Studies discussing the use of a micropulse transscleral laser for the treatment of glaucoma were excluded. A total of 68 full-text articles on MPLT were assessed for eligibility and divided into four sections covering safety, efficacy, and comparisons with conventional laser and intravitreal therapies (Table 1).

## 3. Safety of MPLT

A high safety profile of MPLT was reported in the in vivo and in vitro studies (Table 2). Potential damages were assessed using mathematical models, investigated using animal and stem cell cultures, and measured in imaging tests such as infrared (IR) and red-free fundus photos, optical coherent tomography (OCT), fundus autofluorescence (FAF), microperimetry, fluoresceine angiography (FA), and indocyanine angiography (IGCA).

Ohkoshi et al. [25] detected sites of the application of the micropulse laser in scanning laser ophthalmoscopy in the retro mode. Dark spots were visible immediately after photostimulation, and they were not identified in FAF nor in the fundus photos. However, after 1 week, the alterations were no longer observed. This study implied that MPLT affects the RPE cells and can cause the localized swelling of the treated region.

Luttrull et al. [26] assessed the risk of laser-induced retinal thermal injury by comparing computer modeling of the tissue temperature after MPLT using clinical findings in imaging tests such as IR and red-free fundus photography, FAF, FFA, and OCT. According to the study, an increased risk of retinal damage was related to higher retinal irradiance, and it was found in none of the patients treated with MPLT at a 5% duty cycle.

Wells-Gray et al. [28] confirmed the structural damage after MPLT by measuring the integrity of cone photoreceptors using advanced adaptive optics imaging.

Midena et al., in their studies, pointed to the role of the influence of MPLT on the retinal biomarker levels in aqueous humor [29,30,32]. A strong correlation in protein concentration between the aqueous and vitreous humor was previously proven [37], therefore, a simpler accessible anterior chamber fluid was used for the samples. The authors measured the concentration of the biomarkers of RPE, Müller cells, and a panel of inflammatory molecules in eyes with the DME before and after the MPLT treatment and compared the values with the control groups with healthy ones. The results of their papers were consistent, and they found the effect of MPLT on the expression of aqueous humor markers to be statistically significant. The decrease in proinflammatory proteins and the VEGF level suggested that the MPLT may deactivate the retinal microglia and reduce diabetes-induced inflammation. Moreover, a significant decrease in bioindicators of Müller cell activation implied that MLPT induced positive retinal metabolic and morphology alterations.

Vujosevic et al. [27] showed that both 577 nm and 810 nm micropulse lasers in a “high-density” pattern with 5% DC were safe and efficient in mild DMEs. No retinal damage was detected during any clinical imaging examination. They suggested that the MPLT with the lowest CD and without titration could be a repeatable and simple treatment for patients. In reference to this study, Chang et al. [31] used the same micropulse laser parameters to assess the kinetics of RPE heat-shock protein (HSP) activation. HSP is a group of proteins that are produced in response to cell exposure to stress and during tissue remodeling. This report showed that both the lasers were equally efficient, but a higher predictability and wider safety margin resulted from the use of the 810 nm one. The upregulation of the HSP 70 family was confirmed in the study led by Shiraya et al. [33] on irradiated human RPE stem-cell cultures, which suggested that MPLT could be more beneficial for light perception, photoreceptor protection, and maintenance than a conventional laser could.

In agreement with the results of HSP observation, De Cilla and colleagues [35] proved that MPLT not only reduced oxidative stress and markers of apoptosis, but it also increased autophagia in mouse retinal cells. This study proved that the oxidant–antioxidant balance shifted in favor of the antioxidant system with an increasing number of treatments and with a younger age. Moreover, no laser effect was shown in fellow untreated eyes.

Yu et al. [34] conducted a study on the tissue section of enucleated rabbits’ eyes. In the experiment, the right eyes were treated using an 810 nm micropulse laser, and the left eyes were treated using a 532 nm micropulse laser with 5%, 10%, 20%, and 40% DC. The samples were analyzed for protein marker expression and morphological changes in the retinal tissues. The histologic effect and protein regulation induced by both the lasers were not distinguishable. The 5% DC therapy caused no retinal disruption or RPE damage.

No retinal damage induced by MPLT was confirmed in another animal model investigated by Hirabayashi [36]. According to the upregulation of aquaporin 3 gene expression in retinal photoreceptors, the researchers concluded that MPLT may be responsible for suppressing macular edema and intensifying drainage of retinal fluid. However, the role of aquaporin 3 remains unclear, and it needs to be confirmed in other studies.

## 4. Efficacy of Subthreshold Micropulse Laser and Intravitreal Administration

The efficacy of MPLT was confirmed in various studies (Table 3) as a statistically significant improvement in or stabilization of best corrected visual acuity (BCVA) and a decrease in CRT [14,16,19,38,39,40,41,42,43,44,45,46,47,48,49,50,51,52,53,54]. The first published report on 810 nm micropulsed diode laser was described in 1997 by Friberg et al. [55], and it showed clinical effectiveness in the resolution of a DME. As a limitation of the study, it must be emphasized that OCT devices are not commercially available these days, thus, precise central retinal thickness measurements were not taken.

Nakamura et al. [38] proved that functional improvement after MPLT was limited to an increase in visual acuity. According to the study, the macular sensitivity within the central 10° in microperimetry did not improve significantly, despite the increase in BCVA and the reduction in foveal thickness.

Luttrull et al. [14] observed that significant differences between pre- and postoperative CRT were observed in eyes with CRT < 300 μm, with a maximum reduction between 4 and 7 months after MLPT. The BCVA was stable with a significant improvement between 4 and 7 months of the follow-up.

According to Kwon et al. [51], the MPLT did not cause chorioretinal scars despite repeated treatments occurring and there being an increased number of micropulse shots. The study showed a similar efficacy of the micropulse and conventional lasers.

Inagaki et al. [18] compared the efficacy of 810 nm and 577 nm MPLT combined with focal microaneurysm photocoagulation. They proved that both the wavelengths are effective in reducing CRT and maintaining visual acuity. As advantages of the 577 nm wavelength, they pointed out that it required less power and enabled them to perform both the micropulse and classic therapies using the same device. Supplementary microaneurysm photocoagulation reduced the recurrence rate. Marashi et al. [59] agreed that the hybrid threshold laser of microaneurysms with subthreshold micropulse high-density laser effectively stabilized the DMEs with minimal scar formation.

Mansouri et al. [50] concluded that the retinal thickness affects the spread of the laser energy and influences the tissue response. The authors compared the efficacy of MPLT according to anatomical severity of the edema, suggesting MPLT as an effective and safe therapy in mild and moderate DMEs. In the study, all the eyes with initial CRT > 400 μm did not respond to the therapy and required rescue injections of anti-VEGF. Citirik et al. [56] also showed the relationship between the efficacy of the micropulse laser and the central retinal thickness. The study indicated that eyes which previously underwent ineffective bevacizumab treatment responded well to MPLT if the CRT was no higher than 300 μm.

Nicolò et al. [49] suggested that the micropulse laser is ineffective in eyes which previously did not respond sufficiently to focal or grid macular photocoagulation or an anti-VEGF treatment. Additionally, the authors reported a better response to the treatment of naïve patients, with a stabilization of or improvement in the BCVA and CRT parameters. Valera-Cornejo et al. [41] observed changes in BCVA only in previously untreated patients. It should be underlined that the laser procedures were performed not only over the edema, but also over the entire macula, including the foveal center and unthickened retina. In contrast, the work by Abouhussein et al. [48] led to a different conclusion, i.e., that a single session of MPLT was effective in patients with a refractory DME below 400 μm. In terms of limitations, both the studies had short follow-up and small sample sizes without randomization.

Latalska and colleagues [47] proved that the effects of the micropulse laser were more significant in a rural environment than they were in an urban environment. Moreover, they pointed out that glycated hemoglobin level ≤ 7% significantly influenced the improvement in CRT and near visual acuity.

Optical coherence tomography angiography (OCT-A) is a novel noninvasive accessory examination, which enables imaging vascular abnormalities and microaneurysms in the superficial and deep capillary plexus. It also reveals the enlargement of the foveal avascular zone (FAZ), nonperfused areas, and neovascularization [60]. The studies by Vujosevic et al. [44,45] showed the mechanism of action of a micropulse laser via a reduction in the inflammatory biomarkers detected in OCT and OCT-A. They detected a decreased number of hyper-reflective spots and microaneurysms, whereas the chorioretinal perfusion parameters were stable in response to the MPLT.

No significant changes have been observed in fixed and variable regimens of 577 nm MPLT for mild center-involved DMEs, however, Donati et al. [16] suggested that fixed parameters facilitate the treatment and reduce the number of potential errors. Frizziero et al. [43] confirmed the safety of the fixed model.

Nowacka et al. [58] reported the stabilization of the macular structure through the maintenance of the bioelectrical function of cones and bipolar cells detected in mfERG.

Ueda et al. [40] proved the entropy of RPE cells as an objective indicator of the retinal healing process. They showed a positive correlation between the decrease in CRT after MPLT and entropy measurements in RPE.

According to Işık et al. [39], the response to MPLT may be related to the status of the central RPE and glycated hemoglobin level, however, further studies on a larger group are required.

A recent study by Kikushima et al. [19] compared the 577 nm with the novel 670 nm micropulse treatment. Both the wavelengths seemed to be equally effective, however, the use of the 670 nm laser resulted in less scattering and better penetration.

## 5. Comparison of Subthreshold Micropulse and Conventional Laser Treatment

Studies comparing MPLT with conventional laser therapy are presented in Table 4.

In most reports, the authors found micropulse subthreshold laser therapy to be equivalent to conventional macular photocoagulation [61,62,63,64,65,67,68,69,70,71,72]. Vujosevic et al. pointed out that MPLT is not only as effective as classic lasers are in reducing macular edema, but it is also a less aggressive therapy, as shown by the increased retinal sensitivity in the microperimetry. The positive influence on central retinal sensitivity was also confirmed in the study by Chhlablani et al. [68]. Venkatesh and colleagues [63] suggested that MPLT did not induce any functional loss detected in multifocal electroretinography, with equally good therapeutic effects. Inagaki et al. [64] investigated Japanese patients with a more pigmented retina which could predispose them to the increased absorption of laser energy and more severe retinal damage. Changes in retinal morphology at 3 months after the laser therapy were detected only after pattern scanning and a conventional grid treatment. A recently published multicenter clinical trial by Lois et al. [72] included a large number of participants (266 eyes) with mild DMEs (<400 μm). The study confirmed the clinical effectiveness, safety, and cost-effectiveness of MPLT in compared to those of a conventional laser treatment.

Lavinsky et al. [62] observed the superiority of a high-density, confluent micropulse treatment regarding the anatomical and functional outcomes after 1 year of the follow-up. In contrast, after the normal-density treatment (two burn widths apart), no improvement was seen. Correspondingly, Fazel et al. [67] measured that MPLT significantly improved the BCVA and CRT parameters in eyes with a previously untreated, mild DME. The presented study showed MPLT to be more effective than continuous-wave treatment did in the very short term (4 months). Similarly, Bougatsou et al. [69] agreed that MPLT was more efficacious than a conventional laser was in non-center-involved clinically significant macular edemas, whereas Al-Barky et al. [70] observed slightly better functional outcomes after MPLT. Othman et al. [66] compared MPLT in treatment-naïve patients with MPLT in recurrent or persistent DME 3 months after conventional macular photocoagulation. The therapy was similarly effective in both groups, however, more patients in the secondary group required rescue therapy with an intravitreal steroid.

Available data on alternative subthreshold micropulse panretinal photocoagulation (PRP) in treating severe non-proliferative diabetic retinopathy and proliferative diabetic retinopathy are limited, and without studies of higher quality according to evidence-based medicine (EBM), it should be considered as experimental [73,74].

## 6. Subthreshold Micropulse Laser Treatment and Intravitreal Therapy

Numerous studies compared MPLT with intravitreal treatment or investigated combination therapy (Table 5).

Most articles compared MLPT with bevacizumab, ranibizumab, and aflibercept. The treatment protocol for anti-VEGF monotherapy was three loading injections at a monthly interval followed by a pro re nata (PRN) scheme. The patients qualified for micropulse therapy after receiving three initial loading anti-VEGF doses and with a CRT below 400 µm. It was suggested that additional laser treatment could decrease the burden of agent injection frequency with similar functional and anatomical outcomes [75,78,79,80,82,83,85,86,87]. However, the study by Akhlaghi et al. [77] led to a different conclusion: adjuvant MPLT improved BCVA and CRT in eyes resistant to the bevacizumab therapy.

Inagaki et al. [75] suggested that the initial loading dose of intravitreal anti-VEGF agent, followed by a single MPLT for residual edema reduces the number of required injections and effectively improves BVCA and CRT.

Akkaya et al. [76] proved that MPLT was superior to anti-VEGF injections in patients with mild macular oedema (CRT max. 350 μm) and good visual acuity (BCVA ≤ 0.15 logMAR) due to there being less frequent visits, lower costs, and a higher safety profile. In this regard, MPLT could be considered as the early intervention and, if it is necessary, it can be continued with anti-VEGF injections.

The study by Abdelrahman et al. [81] compared patients treated with MPL or ranibizumab with a control group for multifocal electroretinography (mfERG). The functional outcome was additionally measured not only by the subjective BCVA, but also by objective mfERG readings from the macular region. Only in the ranibizumab group was there a significant improvement in electrophysiological parameters after the treatment. They proved that both MPLT and ranibizumab improved the anatomical and functional retinal parameters, with superiority over the intravitreal agent.

A recent retrospective study by Lai et al. [88] presented that aflibercept monotherapy resulted in short-term higher functional and anatomical improvement compared to that resulting from the MPLT with rescue aflibercept therapy, however, the long-term results did not show any significant differences. In contrast to other studies, MPLT was not preceded by initial anti-VEGF injections, and it was performed with focal conventional laser treatment of microaneurysms.

In general, the authors agree that adjuvant micropulse therapy reduced the number of required intravitreal injections, apart from Koushan et al. [89], who did not find an additional benefit in using a combined therapy.

Elhamid et al. [9] treated center-involved DMEs, which previously did not respond to an anti-VEGF therapy, with a combination of an Ozurdex implant and MLPT. As in other studies, they suggested that poor response after three initial monthly injections of anti-VEGF predicts reduced the persistent response for subsequent doses. An early switch to a steroid implant diminished the number of intravitreal surgeries. In this study, the frequency of recurrence was relatively lower than it was in other trials with the dexamethasone implant, which can be explained by the synergic effect of MPLT. In terms of the limitation, the obtained results require confirmation in larger studies with a control group. Toto et al. [90] also demonstrated the effect of MPLT in addition to a dexamethasone implant. The combined therapy reduced the frequency and the number of required injections, thus extending the treatment-free interval.

Micropulse lasers appear to be an efficient modality to decrease persistent DMEs after pars plana vitrectomy. A comparative study by Bonfiglio et al. [91] showed that MPLT performed 6 months after surgery improved the anatomical and functional parameters in vitrectomized eyes.

## 7. Conclusions

An analysis of the available results is limited due to the scarce number of large, randomized clinical trials. The reviewed studies varied in terms of the inclusion criteria, protocols, and treatment procedures. The detailed eligibility criteria for MPLT have not been defined, however, according to the presented literature, there are some therapeutic principles.

Three meta-analyses which evaluated the efficacy of MPLT versus conventional photocoagulation or intravitreal injections have been published. Chen et al. [92] compared the mean change in BCVA and CRT, according to six randomized controlled trials (RCTs), including a total of 398 eyes. MLPT resulted in better visual acuity with similar anatomical outcome. Similarly, Qiao et al. [93] compared MPLT with an mETDRS treatment in seven RCTs on 425 eyes. They found no statistical differences in BCVA and CRT after the treatments, with less retinal damage after MPLT. Wu et al. [94] performed a Bayesian analysis of 18 studies, comprising a total of 1758 patients, which assessed the effect of lasers in monotherapy or adjuvant therapies to anti-VEGF. The findings showed that ranibizumab plus conventional photocoagulation is more effective than micropulse laser monotherapy is, however, there was no significant difference in efficacy between the MPLT and bevacizumab plus conventional laser treatments, as well as between the MPLT and conventional laser monotherapies.

There are no standardized protocols for MPLT, however, according to the reviewed articles, micropulse panmacular treatment including the fovea, with a fixed regimen, seems to be a cost-effective, noninvasive, and safe therapy. Data in the analyzed articles confirmed that 577 nm laser applications using a 200 μm retinal spot, 200 ms pulse duration, 400 mW power, and 5% DC induced significant morphologic and functional improvement in the central retina and were not associated with any adverse events. Titration can prolong and complicate the procedure. The continuous-wave test burn is performed outside the posterior pole, over non-edematous retina, until a barely visible white spot is created. There is no consensus, after reaching the threshold, on how much to modify the laser power. Some authors switched the continuous wave to the micropulse mode, multiplying the threshold value by 0.5–4. Some researchers titrated the power in micropulse mode and then divided the value by 2. The proper subthreshold value is hard to determine, and medical errors can lead to overtreatment and involuntary damage of the retina. A confluent treatment using fixed 400 mW power for yellow laser with low 5% DC and high intensity was confirmed to effectively stimulate RPE cells.

None of the presented studies detected any visible signs of chorioretinal damage in the ancillary imaging tests and animal retinal sections. In contrast to harmful conventional laser treatment, MPLT additionally increased the central retinal sensitivity.

The efficacy of the micropulse laser was proven in mild DMEs with a CRT that is smaller than 400 μm due to the diffused distribution in the target tissue. In general, the treatment helps to stabilize or improve the visual acuity and decrease the macular edema. Better results are observed in a high-density protocol covering the macular region, with no spacing between the spots. Automatic pattern systems are helpful in the application of invisible laser spots. The minimal interval from the treatment to obtain a significant response and a reduction in retinal thickness is about 3 months. Therefore, it can be recommended to start the therapy with three loading doses of anti-VEGF, followed by MPLT combined with PRN injections to achieve a quick response to anti-VEGF, which is supported by the long-lasting remodeling effect of MPLT. The increased number of micropulse sessions is associated with a greater retinal response. A combined treatment requires a lower number of anti-VEGF injections, and it is not inferior to monotherapy [95,96]. MPLT is also an emerging option as a standalone treatment for noncompliant patients and for those having contraindications for other therapies.

## Figures and Tables

**Figure 1 jcm-12-00274-f001:**
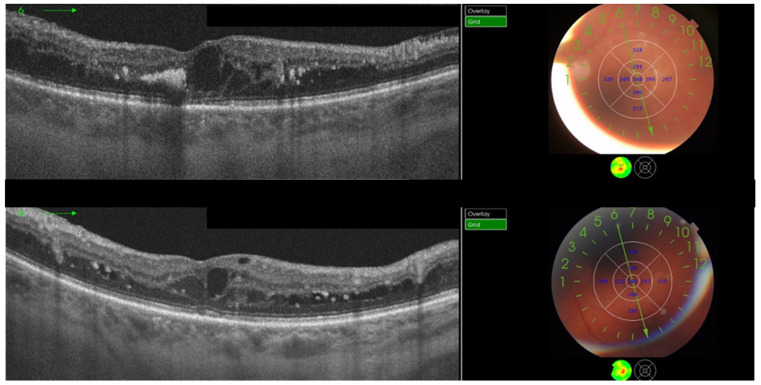
A patient with diabetic macular edema, showing optical coherence tomography at baseline (**top**) and 3 months after subthreshold micropulse laser treatment (**bottom**). Treatment consisted of 510 confluent panmacular applications using a 200 μm retinal spot, 200 ms pulse duration, 400 mW power, and 5% duty cycle. The central retinal thickness was equal (348 μm) at baseline and at 3 months and the mean retinal thickness decreased.

**Table 1 jcm-12-00274-t001:** Search strategy.

Date of search	June 2022–November 2022
Databases searched	PubMed Mendeley
Target items	Journal papers
Years covered by search	2010–2022
Language	English
Search terms used	(subthreshold laser OR micropulse laser) NOT transscleral AND (diabetic macular edema OR clinically significant macular edema) AND (anti-VEGF OR intravitreal steroid OR vitrectomy) AND (conventional photocoagulation OR ETDRS photocoagulation OR continuous-wave photocoagulation) AND combined therapy AND safety

**Table 2 jcm-12-00274-t002:** Studies on the safety of MPLT.

Article	Design	Evaluated on	Results
**Clinical Study**			
Ohkoshi et al., 2010 [25]	Interventional case series 810 nm MPLT 125–200 μm/200 ms/15% DC titrated to 2–3× CWL power	8 eyes CS-DME; 1 eye BRVO	Dark spots in SLO (retro mode) were observed with a higher laser energy and could be related to the swelling of RPE. No laser scars on color fundus images and FAF.
Luttrull et al., 2012 [26]	Retrospective 810 nm MPLT computer modeling of tissue temperature 125 μm/300 ms/5% DC fixed 950 mW	212 eyes DME; 40 eyes BRVO FU = 12	The risk of thermal retinal damage was low and could be eliminated by 5% DC. Significant decreases in CRT and maximum macular thickness.
Vujosevic et al., 2015 [27]	Prospective, randomized 577 nm (100 μm/200 ms/5% DC/fixed 250 mW) vs. 810 nm (125 μm/200 ms/5% DC/fixed 750 mW) MPLT	53 eyes untreated CI-DME ≤ 400 μm FU = 6	No visible scars on FA and FAF in both groups. No statistically differences in CRT, BCVA, and microperimetry in both groups. Statistically improved macular sensitivity in both groups.
Wells-Gray et al., 2018 [28]	Observational 577 nm MPLT Evaluation of the integrity of cones 200 μm/200 ms/5% DC fixed 400 mW, 7 × 7 pattern	4 eyes non-CI-DME	No signs of structural photoreceptors damage in adaptive optics imaging.
Midena et al., 2019 [29]	Prospective 577 nm MPLT AH concentration of RPE biomarkers 200 ms/5% DC/fixed 250 mW	18 eyes naïve DME ≤ 400 μm; 10 healthy eyes FU = 12	Significant difference in RPE biomarkers concentration in DME and healthy patients. No significant influence of MLPT on biomarkers.
Midena et al., 2019 [30]	Prospective 577 nm MPLT AH concentration of 58 inflammatory biomarkers 200 ms/5% DC/fixed 250 mW	18 eyes naïve DME ≤ 400 μm; 10 healthy eyes FU = 12	Significant decrease in proinflammatory cytokines produced by microglial cells after MPLT.
Chang et al., 2020 [31]	577 nm vs. 810 nm MPLT Mathematical analysis of the kinetics of HSP activation Vujosevic et al. [27] laser parameters	n/a	Both lasers were equally effective. The 810 nm laser had a significantly wider safety margin.
Midena et al., 2020 [32]	Prospective 577 nm MPLT AH concentration of Müller cells biomarkers 200 ms/5% DC/fixed 250 mW	10 eyes naïve DME ≤ 400 μm; 12 healthy eyes FU = 12	Significant reduction in concentration of VEGF and Müller cell activation markers in AH after MPLT. Significantly higher level of biomarkers in AH in patients with DME.
Shiraya et al., 2022 [33]	577 nm MPLT Transcriptome analysis using RNA sequencing MPLT vs. conventional 100 μm/200 ms/5% DC 100–300 mW, 5 × 5 pattern	Human pluripotent stem cell-derived RPE cells FU = 24 h	MPLT induced the expression of HSP and differentially expressed genes related to photoreceptors.
**Animal study**			
Yu et al., 2013 [34]	Prospective 532 nm (130 μm/200 ms) vs. 810 nm (195 μm/300 ms) MPLT titrated to 1× CWL power DC = 5%, 10%, 20%, 40%	14 rabbits FU = 1	No significant differences in histologic changes and protein expression between 532 and 810 nm laser. Lower DCs were more selective and caused less retinal photocoagulation
De Cilla et al., 2019 [35]	Prospective 810 nm Changes in apoptotic proteins expression 500 μm/75 ms/5% DC titrated to 2× CWL power	20 young mice and 20 old mice one treatment (*n* = 20) and three treatments (*n* = 20) FU = 6	MPLT modulated balance between oxidant and antioxidant in retina; regulated activation of apoptosis and autophagia.
Hirabayashi et al., 2022 [36]	Prospective 577 nm MPLT RT-PCR analysis of retinal samples of the expression of aquaporin subtypes and angiogenesis-related factors 50 μm/150 ms/5% DC/fixed 50 mW	Mice Threshold (*n* = 6) MPLT (*n* = 6) control (*n* = 6) FU = 3 days	Elevated aquaporin 3 expression in MPLT in compared to control group.

MPLT, micropulse laser treatment; CWL, continuous-wave laser; CS, clinically significant; DME, diabetic macular edema; SLO, scanning laser ophthalmoscopy; FU, follow-up (in months); DC, duty cycle; CRT, central retinal thickness; CI, center involved; FA, fluoresceine angiography; FAF, fundus autofluorescence; BCVA, best corrected visual acuity; AH, aqueous humor; RPE, retinal pigment epithelium; HSP, heat-shock protein; VEGF, anti-vascular endothelial growth factor.

**Table 3 jcm-12-00274-t003:** Studies on efficacy of MPLT in DME treatment.

Article	Design	Evaluated on	Results
Nakamura et al., 2010 [38]	Prospective 810 nm MPLT 200 μm/200 ms/15% DC titrated to 2× CWL power, without papillomacular bundle	28 eyes diffuse DME FU = 3	Significant improvement in BCVA; significant reduction in CRT (from 481 ± 110 to 388 ± 127 μm). Central retinal sensitivity did not improve significantly.
Ohkoshi et al., 2010 [53]	Prospective 810 nm MPLT 200 μm/200–300 ms/15% DC titrated to 2× CWL power	43 eyes CS-DME < 600 μm FU = 3	BCVA did not change significantly. Significant reduction in CRT (from 341.8 ± 119.0 to 289.5 ± 122.8 μm).
Takatsuna et al., 2011 [52]	Retrospective 810 nm MPLT 200 μm/100 ms/15% DC titrated to 2× CWL power	56 eyes DME FU = 12	BCVA did not change significantly. Significant reduction in CRT (from 504.3 ± 105.8 to 320.4 ± 134.9 μm).
Luttrull et al., 2014 [14]	Retrospective 810 nm MPLT 125–200 μm/300 ms/5% DC fixed 780 mW or 950 mW	39 eyes CI-DME, V > 20/40 FU = 12	Significant improvement in BCVA (logMAR from 0.19 ± 0.11 to 0.16 ± 0.09); significant reduction in CRT for eyes with CRT < 300 μm (from 248.3 ± 27.8 to 229.4 ± 34.3 μm). No evidence of MPLT injury to RPE.
Kwon et al., 2014 [51]	Retrospective 577 nm MPLT 100 μm/20 ms/15% DC titrated to immediately below CWL power 3 × 3 pattern, 1.5 widths	14 eyes DME > 260 μm FU = 8	Significant improvement of BCVA (logMAR from 0.51 ± 0.42 to 0.40 ± 0.35). Nonsignificant decrease in CRT. No laser scars detected in color photographs, FAF, IR, and FA.
Mansouri et al., 2014 [50]	Retrospective 810 nm MPLT CRT ≤ 400 μm vs. CRT > 400 μm 125 μm/300 ms/5% DC fixed 950 mW	63 eyes DME FU = 12	Significant reduction in CRT and gain in BCVA in patients with CRT ≤ 400 μm, stable CRT and BCVA in patients with CRT > 400 μm. No adverse effect from MPLT.
Nicolò et al., 2014 [49]	Retrospective 577 nm MPLT Naïve vs. previously treated DME 200 μm/200 ms/5% DC fixed 200 mW	22 eyes DME FU = 6	Significant improvement in BCVA (logMAR from 0.39 ± 0.19 to 0.27 ± 0.17) and CRT (from 350.9 ± 74.7 to 311.2 ± 49.43 μm) only in naïve patients. No evidence of RPE damage in FAF.
Inagaki et al., 2015 [18]	Prospective 810 nm MPLT + 561 nm focal laser of microaneurysms vs. 577 nm MPLT + 577 nm focal laser of microaneurysms 200 μm/200 ms/15% DC titrated to 2× CWL power, up to 500 μm from fovea	53 eyes CS-DME FU = 12	Similar significant reduction in CRT on both groups. Stable BCVA in both groups. Retreatment rate higher in 810 nm group (16.7% vs. 3.4%). Mean power lower in 577 nm group (204.1 vs. 954.1 mW).
Abouhussein et al., 2016 [48]	Prospective 577 nm MPLT previously treated 200 μm/200 ms/5% DC fixed 400 mW	20 eyes CI-DME ≤ 400 μm FU = 6	Significant improvement in BCVA (logMAR from 0.42 ± 0.15 to 0.3 ± 0.26), significant reduction in CRT (from 354.3 ± 32.96 to 310.7 ± 52.62 μm). No evidence of retinal scars in fundus photography or FA.
Latalska et al., 2017 [47]	Prospective 577 nm MPLT Rural vs. urban patients 100 μm/20 ms/5% DC titrated to 2× CWL power	75 eyes Diffuse DME FU = 6	More significant treatment effects in rural patients. Significant improvement of reading visual acuity and decrease in CRT in both groups. BVCA remained stable. No retinal damage.
Değirmenci et al., 2018 [54]	Retrospective 577 nm MPLT 160 μm/200 ms/5% DC titrated to 0.5× visible MPLT power	9 eyes non-FI-DME FU = 3	Significant decrease in mean retinal thickness (from 470.6 to 416 μm). Nonsignificant improvement of BCVA. No evidence of laser scars in FAF.
Vesela et al., 2018 [46]	Retrospective 577 nm MLPT 160 μm/200 ms/5% DC titrated to 0.3–0.5× CWL power	63 eyes DME FU = 12	Significant decrease in CRT (from 442 to 379 μm). Stabilization of BCVA.
Citirik et al., 2019 [56]	Prospective 577 nm MPLT CRT 250–300 μm vs. 301–400 μm vs. > 400 μm vs. healthy control 160 μm/200 ms/5% DC titrated to 0.5× visible MPLT power	80 eyes recurrent DME after injection FU = 6	Significant reduction in CRT (from 276.0 ± 22.44 to 238.57 ± 25.87 μm) and gain in BCVA (logMAR from 0.52 ± 0.05 to 0.38 ± 0.04) only in patients with pretreatment CRT ≤ 300 μm.
Vujosevic et al., 2020 [44,45]	Prospective 577 nm MLPT vs. control 100 μm/200 ms/5% DC fixed 250 mW, 7 × 7 pattern	52 eyes naïve DME ≤ 400 μm FU = 12	Significant increase in BCVA (ETDRS score from 69.4 ± 12.0 to 76.0 ± 9.1). Significantly decrease in hyper-reflective retinal spots, microaneurysms, DRIL. CRT did not change. Stable parameters in control group. No need for rescue treatment. No changes in FAF.
Donati et al., 2021 [16]	Retrospective 577 nm MPLT fixed vs. variable treatment regimen 100 μm/200 ms/5% DC fixed 450 mW or titrated to 4× CWL power	39 eyes DME < 400 μm FU = 12	Equally significant decrease in CRT in both groups. No significant improvement in BCVA in both groups.
Frizziero et al., 2021 [57]	Retrospective 577 nm MPLT 100 μm/200 ms/5% DC/fixed 250 mW	134 eyes naïve CI-DME CRT ≤ 400 μm FU = 12	Significant improvement in BCVA (EDTRS score from 77.3 ± 4.5 to 79.4 ± 4.4). No significant CRT reduction. No adverse effects in FAF and OCT.
Kikushima et al., 2021 [19]	Retrospective 577 nm vs. 670 nm MPLT 200 μm/200 ms/10% DC titrated to immediately below CWL power	43 eyes DME FU = 1	Both lasers maintained BCVA. CRT equally significantly decreased in both groups. No changes in FAF.
Nowacka et al., 2021 [58]	Prospective 577 nm MPLT 5% DC, titrated, not exceeding 350 mW	21 eyes CI-DME CRT < 400 μm FU = 6	No significant change in BCVA, CRT, bioelectrical function of cones, and bipolar cells in mfERG.
Passos et al., 2021 [42]	Retrospective 577 nm MPLT 160 μm/200 ms/5% DC titrated to 0.5× CWL power	56 eyes CI-DME FU = 3	Significant improvement in BCVA (logMAR from 0.59 ± 0.32 to 0.43 ± 0.25). Different OCT instruments disabled CRT analysis.
Ueda et al., 2021 [40]	Prospective 577 nm MPLT Evaluation of the dynamics of retinal healing process 100 μm/200 ms/5% DC titrated to 0.5× CWL power, 7 × 7 pattern	11 eyes DME FU = 6	Decrease in RPE entropy after MPLT on polarization-sensitive OCT. No visible signs in color photography, FAF, and OCT. No significant changes in BCVA and CRT.
Valera-Cornejo et al., 2021 [41]	Prospective 577 nm MPLT naïve vs. previously treated DME 100–150 μm/200 ms/5% DC titrated to 0.5× CWL power, 8 × 8 pattern	33 eyes CI-DME < 700 μm FU = 3	No significant changes in BCVA for both groups. Significant reduction in CRT (from 420 ± 121 to 390 ± 130 μm) in naïve group. No adverse events in color photographs and FAF.
Işık et al., 2022 [39]	Retrospective 577 nm MPLT MPLT vs. healthy control 160 μm/200 ms/5% DC titrated to 0.5× visible MPLT power	40 eyes CI-DME FU = 3	Significant increase in BCVA; significant decrease in CRT. Area of central RPE measured in EDI-OCT was smaller in patients requiring retreatment.
Marashi et al., 2022 [59]	Retrospective 532 nm focal threshold laser of microaneurysms + 532 nm grid MPLT 125 μm/200 ms/5% DC titrated to 0.5× CWL power	12 eyes DME CRT > 300 μm FU = 6	Significant reduction in CRT (from 336.58 ± 86.36 to 264.33 ± 61.41 μm). Stable BCVA. Minimal scar formation. Four eyes required anti-VEGF injection.

MPLT, micropulse laser treatment; CWL, continuous-wave laser; DME, diabetic macular edema; FU, follow-up (in months); BCVA, best corrected visual acuity; CRT, central retinal thickness; CS, clinically significant; CI, center involved; RPE, retinal pigment epithelium; FAF, fundus autofluorescence; IR, infrared; FA, fluoresceine angiography; DRIL, disorganization of inner retinal layers; mfERG, multifocal electroretinography; EDI, enhanced-depth imaging.

**Table 4 jcm-12-00274-t004:** Studies on MPLT and conventional laser.

Article	Design	Evaluated on	Results
Vujosevic et al., 2010 [61]	Prospective, randomized 810 nm MPLT (125 μm/200 ms/5% DC/fixed 750 mW) vs. 514 nm mETDRS photocoagulation (100 μm/100 ms/80–100 mW)	62 eyes naïve CI-DME CRT ≥ 250 μm FU = 12	Stable BCVA in both groups. Similarly significant decrease in CRT in both groups. Significant increase in central 12° retinal sensitivity in MPLT group and significant decrease in ETDRS group. Mean no. of treatment 2.03 ± 0.75 in MPLT vs. 2.1 ± 1 in mETDRS.
Lavinsky et al., 2011 [62]	Prospective, randomized, double-masked 810 nm ND-MPLT vs. 810 nm HD-MPLT (125 μm/300 ms/15% DC/titrated to 1.2× CWL power) vs. 532 nm mETDRS photocoagulation (75 μm/50 ms/barely visible)	123 patients naïve CS-DME CRT ≥ 250 μm FU = 12	Best improvement in BCVA in HD-MPLT group (logMAR 0.25); stable BCVA in ND-MPLT group. Significant progressive reduction in CRT in all groups; greatest in HD-MPLT group (154 μm). No statistical differences in BCVA and CRT in HD-MPLT and mETDRS group. No retreatment in 49% HD-MPLT, 44% mETDRS, and 2% ND-MPLT.
Venkatesh et al., 2011 [63]	Prospective, randomized 810 nm MPLT (125 μm/2000 ms/5% DC/titrated to 0.5× CWL power) vs. 532 nm conventional laser (50–100 μm/100 ms/90–180 mW)	46 eyes CS-DME CRT < 400 μm FU = 6	Stable BCVA, macular sensitivity, and contrast sensitivity in both groups. Similarly significant decrease in CRT in both groups. More regions with functional loss in mfERG detected after conventional laser.
Inagaki et al., 2012 [64]	Retrospective, case-series Grid photocoagulation 810 nm MPLT (200 μm/200 ms/15% DC, titrated to 2–3× CWL power) vs. multicolor (532, 561, or 569 nm) laser (100 μm, 100 ms, 50–100 mW) vs. 532 nm pattern scanning laser (100 μm, 20 ms, 120–320 mW)	30 eyes CS-DME (*n* = 15) BRVO (*n* = 15) FU = 6	No damage was identified after MPLT in OCT scans; fewer changes in outer retina after pattern scanning laser than after conventional laser.
Xie et al., 2013 [65]	Prospective, randomized 810 nm MPLT (125 μm/300 ms/5% DC, titrated to 0.5× visible MPLT power) vs. argon ion conventional laser	99 eyes DME FU = 6	Stable BCVA and significant decrease in CRT in both groups. No significant differences in BCVA and CRT between groups.
Othman et al., 2014 [66]	Prospective 810 nm MPLT (75–125 μm/15% DC/800–1000 mW) Primary treatment vs. secondary treatment after argon laser photocoagulation	220 eyes CS-DME CRT > 210 μm FU = 14 ± 2.8	Stable BCVA in both groups. Significant decrease in CRT in both groups. In primary treatment, 11.37% of eyes and 33% of them in secondary treatment required intravitreal triamcinolone; 3.2% of them in primary group required vitrectomy due to poor response.
Fazel et al., 2016 [67]	Prospective, randomized, single-blind 810 nm MPLT (75–125 μm/0.3 ms/15% DC/1000 mJ) vs. 810 nm focal + grid conventional laser	68 eyes naïve CS-DME CRT 300–450 μm FU = 4	Significant improvement in BCVA only in MPLT group. Significant decrease in CRT in both groups; more significant in MPLT group.
Chhablani et al., 2018 [68]	Prospective, randomized, double-masked 577 nm 5% DC vs. 577 nm 15% DC (100 μm/100 ms/titrated to 0.3× visible MPLT power) vs. 532 nm navigated mETDRS photocoagulation.	30 eyes naïve non-CI-DME CRT < 350 μm FU = 3	Stable BCVA and CRT in all groups. Significant reduction in retinal sensitivity in conventional group; similarly significant increase in 5% DC and 15% DC groups.
Bougatsou et al., 2020 [69]	Prospective, randomized 532 nm MPLT (50–100 μm/100 ms/15% DC/titrated to 2× CWL power) vs. 532 nm focal photocoagulation	60 eyes non-CI-CS-DME naïve FU = 6	Significantly reduced CRT in both groups; significantly better in MPLT group. Significant improvement in BCVA in MLPT group.
Al-Barki et al., 2021 [70]	Prospective short-pulse subthreshold 532 nm vs. micropulse 810 nm	116 eyes CI-DME FU = 6	Visual acuity significantly improved in MLPT group. Comparable anatomic results and need for rescue therapy in both groups.
Lois et al., 2022 [71,72]	Prospective, randomized, double-masked DIAMONDS trial 577 nm MPLT (125 μm/300 ms, 15% DC, titrated to 1× CWL power) vs. mETDRS photocoagulation	266 eyes CI-DME CRT 301–399 μm FU = 24	No difference in BCVA, CRT, or 10-2 visual field; need for additional rescue treatment. No. of laser treatments was higher in MPLT group (2.37 vs. 1.89).

MPLT, micropulse laser treatment; mETDRS, modified Early Treatment Diabetic Retinopathy Study; CWL, continuous-wave laser; CI, center involved; DME, diabetic macular edema; BCVA, best corrected visual acuity; CRT, central retinal thickness; FU, follow-up (in months); ND, normal density; HD, high density; CS, clinically significant; mfERG, multifocal electroretinography; DC, duty cycle.

**Table 5 jcm-12-00274-t005:** MPLT and intravitreal therapy.

Article	Design	Evaluated on	Results
Inagaki et al., 2019 [75]	Retrospective ranibizumab or aflibercept + 577 nm MPLT 100 μm/200 ms/5% DC titrated to 0.5–0.6× CWL power	34 eyes DME FU = 12	Significant improvement in BCVA (logMAR from 0.52 ± 0.34 to 0.43 ± 0.33). Significant decrease in CRT (from 491.1 to 354.8 μm). Mean no. of injections 3.6 ± 2.1.
Akkaya et al., 2020 [76]	Retrospective ranibizumab or aflibercept vs. 577 nm MPLT 100 μm/200 ms/10% DC titrated to 0.5× CWL power	76 eyes CI-DMI ≤ 350 μm BCVA > 0.7 Snellen FU = 12	BVCA significantly better in laser group (logMAR 0.054 ± 0.07 vs. 0.095 ± 0.08). The decrease in CRT was non-significant, but it was higher in laser group. Mean no. of injections 5.85 ± 1.38; mean no. of laser treatments 3.64 ± 0.76.
**Bevacizumab**			
Akhlaghi et al., 2019 [77]	Prospective, randomized bevacizumab + 532 nm MPLT vs. bevacizumab 200 μm/5% DC titrated to 4× CWL power	42 eyes refractory DME FU = 3	Significant improvement in BCVA (logMAR 0.81 ± 0.33 to 0.62 ± 0.26) and significant decrease in CRT (from 513 ± 126.29 to 408.1 ± 95.28 μm) only in combination group.
Altınel et al., 2021 [78]	Retrospective bevacizumab + 577 nm MPLT vs. bevacizumab 160 μm/200 ms/5% DC titrated to 0.5 × visible MPLT power	80 eyes CI-DME FU = 12	Significant increase in BVCA in combined group. Significant decrease in CRT, which was similar in both groups. Mean no. of injections significantly lower in combined group (4.38 ± 0.81 vs. 5.65 ± 1.51).
El Matri et al., 2021 [79]	Retrospective bevacizumab + 577 nm MPLT vs. bevacizumab 200 μm/200 ms/5% DC fixed 400 mW, 2 × 2 or 4 × 4 pattern	98 eyes naïve CI-DME ≤ 500 μm FU = 12	Significant improvement in BCVA (logMAR from 0.692 ± 0.35 to 0.501 ± 0.37) and decrease in CRT (from 479.1 ± 14.3 to 289.6 ± 15) in combined group. The difference is not significant between groups. Significantly lower no. of injections in combined group (4.1 ± 1.5 vs. 7.2 ± 1.3) per year.
**Ranibizumab**			
Moisseiev et al., 2018 [80]	Retrospective 577 nm MPLT vs. ranibizumab 200 μm/200 ms/5% DC/fixed 400 mW	38 eyes DME FU = 12	Comparable improvement in BCVA. Change in CRT greater in ranibizumab group. Significantly fewer injections required in MPLT group (1.7 ± 2.3 vs. 5.6 ± 2.1).
Abdelrahman et al., 2020 [81]	Prospective, randomized 532 nm MPLT vs. ranibizumab vs. control 200 μm/200 ms/5% DC fixed 400 mW, 7 × 7 pattern	120 eyes naïve DME CRT ≤ 400 μm FU = 6	Significant improvement in BCVA (93% vs. 31%) and decrease in CRT (34.66% vs. 11.69%) in both groups; significantly higher in ranibizumab group. Significant improvement in mfERG only in ranibizumab group.
Furashova et al., 2020 [82]	Prospective, randomized ReCaLL clinical trial 810 nm MPLT 200 ms/15% DC titrated to 2× CWL power, without fovea	17 eyes DME CRT > 300 μm FU = 12	Significant increase in BCVA with significant decrease in CRT in both groups. No significant differences between groups. Significantly lower no. of injections in combined group (7.5 vs. 9).
Bıçak et al., 2022 [83]	Retrospective ranibizumab + 577 nm MPLT vs. ranibizumab 165 μm/200 ms/5% DC titrated to 0.5× visible MPLT power, grid pattern	97 eyes DME CRT ≤ 350 μm FU = 9	Significant increase in BCVA in both groups with significant decrease in CRT. No significant differences between groups. Significantly lower no. of injections in combined group (4.19 ± 1.01 vs. 5.53 ± 1.14).
Mi et al., 2022 [84]	Prospective, randomized, double-blind ranibizumab + sham 577 nm MPLT vs. sham ranibizumab + 577 nm MPLT 200 μm/200 ms/5% DC fixed 400 mW, 7 × 7 pattern	72 patients DME CRT > 300 μm	This study is currently recruiting participants. The results are not yet available.
**Aflibercept**			
Khattab et al., 2019 [85]	Prospective, randomized aflibercept + 577 nm MPLT vs. aflibercept 200 μm/200 ms/5% DC fixed 400 mW, 7 × 7 pattern	54 eyes DME CRT > 250 μm FU = 18	Significant increase in BCVA and contrast sensitivity; significant decrease in CRT in both groups. No significant differences between groups. Significantly lower no. of injections in combined group (4.1 ± 1.1 vs. 7.3 ± 1.1).
Abouhussein et al., 2020 [86]	Prospective, randomized aflibercept + 577 nm MPLT vs. aflibercept 200 μm/200 ms/5% DC fixed 400 mW, 5 × 5 pattern	40 eyes naïve DME CRT > 300 μm FU = 12	Significant increase in BCVA and significant decrease in CRT in both groups. No significant differences between groups. Significantly lower no. of injections in combined group (4.5 ± 1.4 vs. 5.4 ± 1.7) after the loading dose of aflibercept.
Kanar et al., 2020 [87]	Prospective, randomized aflibercept + 577 nm MPLT vs. aflibercept 160 μm/200 ms/5% DC titrated to 0.5× visible MPLT power	56 eyes naïve DME CRT > 300 μm FU = 12	Significant increase in BCVA and significant decrease in CRT in both groups. No significant differences between groups. Significantly lower no. of injections in combined group (3.21 ± 0.41 vs. 5.39 ± 1.54).
Lai et al., 2021 [88]	Retrospective 577 nm MPLT + focal laser of microaneurysms vs. aflibercept 200 μm/200 ms/5% DC fixed 400 mW, 5 × 5 pattern, 0.25 spacing	164 eyes DME CRT > 300 μm FU = 24	Significant increase in BCVA and significant decrease in CRT in both groups. Significantly greater improvement in BCVA at 6 months, as well as in CTR at 6 and 12 months in aflibercept group; no significant differences between groups at 12 and 24 months. Rescue aflibercept required in 24% of MLPT eyes.
Koushan et al., 2022 [89]	Prospective, randomized, single-blind DAM Study aflibercept + 532 nm MPLT vs. aflibercept + sham 532 nm MPLT 200 μm/200 ms/10% DC titrated to 0.9× visible MPLT power, 3 × 3 pattern	30 eyes CI-DME CRT ≥ 315 μm FU = 12	Significant increase in BCVA and significant decrease in CRT in both groups. No significant differences between groups. Similar no. of injections in both groups.
**Dexamethasone**			
Elhamid 2017 [9]	Prospective 577 nm MPLT + IDI 200 μm/200 ms/5% DC fixed 400 mW, 7 × 7 pattern	20 eyes refractory CI-DME CRT ≥ 300 μm FU = 12	Significant improvement in BCVA (Snellen from 0.45 ± 0.14 to 0.6 ± 0.1) and significant decrease in CRT (from 420.7 ± 38.74 to 285.2 ± 14.99 μm). Retreatment was performed in 40% of eyes.
Toto et al., 2022 [90]	Prospective 577 nm navigated MPLT + IDI vs. IDI 100 μm/100 ms/5% DC titrated to 0.3× visible MPLT power	60 eyes naïve CI-DME CRT > 300 μm FU = 6	Significant improvement in BCVA and decrease in CRT in both groups; significantly higher in MPLT+ IDI group. Significantly higher no. of second injections in IDI group (73.3% vs. 56.7% of patients). Shorter time before the second injection in IDI group (83.5 vs. 137.4 days).
**Vitrectomy**			
Bonfiglio et al., 2022 [91]	Prospective PPV+ 577 nm MPLT vs. PPV 200 μm/200 ms/5% DC titrated to 2× CWL power, 7 × 7 pattern	95 eyes Persistent DME CRT ≥ 300 μm FU = 6	Significant improvement in BCVA (EDTRS letters from 51.54 ± 13.81 to 57.83 ± 13.95) and decrease in CRT (from 410.59 ± 129.91 to 283.39 ± 73.45 μm) in MPLT group. Second MPLT required in 67% of eyes. Parafoveal VD significantly higher and FAZ significantly smaller in OCTA in MPLT group.

MPLT, micropulse laser treatment; CWL, continuous-wave laser; DME, diabetic macular edema; FU, follow-up (in months); BCVA, best corrected visual acuity; CRT, central retinal thickness; CI, center involved; mfERG, multifocal electroretinography; IDI, intravitreal dexamethasone implant; PPV, pars plana vitrectomy; FAZ, foveal avascular zone; VD, vessel density; OCTA, optical coherent tomography angiography.

## Data Availability

Not applicable.

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
