# Peer review of "Subthreshold Micropulse Laser for Diabetic Macular Edema: A Review"

_jcm, 2022, doi:10.3390/jcm12010274_

Round 1

Reviewer 1 Report

This is well written review on MPLT for DME.

(1) As for safety of MPLT, it is better to divide into two categories, animal study and clinical study. 

(2) In Figure 1, please indicated the CRT after treatment or show the map or macular volume before and after treatment. 

(3) There are only 2 paper about PRP using MPLT, it would be clear to delete these sections.

Author Response

Thank you for your valuable comments.

Response 1: Table 2 was revised and divided into 2 parts: clinical and animal study. The order of the paragraphs describing MPLT safety has changed, clinical studies was described first, followed by animal studies.

Response 2: Figure 1 was improved: the macular volume map was added and a word "stable" was replaced by "equal" (baseline CRT and after treatment did not change).

Response 3: Sections on PRP were deleted. A short paragraph briefly describing scarceness of data on MPLT in PRP was written.

Additionally, in Table 4 missing follow-up periods were added. 

I uploaded the revised manuscript instead of the cover letter and I cannot delete the wrong file. I cannot use "Track Changes" function because it did not work with Mendeley Cite, changes are in red. 

Reviewer 2 Report

In the field of  subthreshold micropulse laser treatment of diabetic macular edema, the data can not be very precise and accurate because neither the two entities can be precisely known; the mechanism of cellular and sub cellular action of micropulse laser on the one hand and the biohumoral profile of each patient on the other.

Author Response

Thank you for the valuable comment. We agree that there is still much to be explored in the field of micropulse laser therapy and diabetic macular edema.